Energy conservation prospects in water intensive Paddy-Wheat cropping system for groundwater pumping in the semi-arid region of Haryana

http://orcid.org/0000-0002-4119-5997 Singh Kuldeep 1 dkuldeepv@gmail.com
Jhorar Raj Kumar 2
Sidhpuria Manohar Sahai 2
Kumar Mukesh 2
http://orcid.org/0000-0003-0948-4876 Mehla Mukesh Kumar 3
1 Krishi Vigyan Kendra, Chaudhary Charan Singh Haryana Agricultural University , Sonipat, Haryana , India
2 College of Agricultural Engineering and Technology, Chaudhary Charan Singh Haryana Agricultural University , Hisar, Haryana , India
3 Department of Soil and Water Engineering, College of Technology and Engineering, Maharana Pratap University of Agriculture and Technology , Udaipur, Rajasthan , India
Meraj Gowhar
Electronic publication date: 2023 Jan 26
Publication date: 2023
Volume: 11
Electronic Location ID: e14815
Received 2022 Sep 19; Accepted 2023 Jan 6
Copyright: © 2023 Singh et al.
Copyright year: 2023
Copyright holder: Singh et al.
License: This is an open access article distributed under the terms of the Creative Commons Attribution License, which permits unrestricted use, distribution, reproduction and adaptation in any medium and for any purpose provided that it is properly attributed. For attribution, the original author(s), title, publication source (PeerJ) and either DOI or URL of the article must be cited.
License URL: https://creativecommons.org/licenses/by/4.0/

Keywords: Groundwater pumping, Energy conservation, Energy saving, Pumping efficiency, Irrigation

Funding: The authors received no funding for this work The authors received no funding for this work.

==============================
The study was aimed at identifying the potential energy saving for groundwater pumping through enhanced efficiency of the pump sets. A total of 65 electrically powered tube wells were selected in the Sonipat district of the north Indian State of Haryana to study the energy use efficiency for groundwater pumping. The existing efficiency as well as the minimum expected overall efficiency of the tubewells were determined based on field survey, measurements, and applicable standard code of the Bureau of India Standards. The overall efficiencies of selected tube wells, computed based on actual measured power consumption, varied from 10.1% to 56.6% in the Sonipat block and 15.3% to 52.8% in the Rai block. The average energy requirement, for the selected tube wells, at the current efficiency level was 4,364.0 and 13,100.4 kWh for wheat and paddy crops, respectively, in the Sonipat block, while it was 3,424.8 and 10,280.9 kWh for wheat and paddy crops, respectively, in the Rai block. Analysis revealed that improving overall efficiency from the current level to the minimum expected level can lead to energy savings of 48.3% and 35.9% for tube wells in the Sonipat and Rai block, respectively. In the Rai block, where the groundwater level has declined significantly, the replacement of inefficient pumps should be done in tandem with crop diversification, improving water application efficiency and groundwater status by employing improved irrigation management practices and adopting groundwater recharge techniques.

Introduction

Water and energy scarcity are well-known and challenging issues for the long-term sustainability of natural resources around the world. Water and energy are inevitably intertwined natural resources (Schnoor, 2011). It is necessary to build efficient water and energy systems for long-term use (Siddiqi & Anadon, 2011). As of late, groundwater irrigation has prospered as a key resource for an assured supply of water to farmers (Singh, Kasana & Bhardwaj, 2022). As a result, the groundwater irrigated area in India increased from 12 million ha to 40 million ha between 1970 and 2010 (Ministry of Statistics and Programme Implementation, Government of India (MoSPI), 2015). The extraction of groundwater from wells consumes about half of the electricity utilized by India’s agriculture sector (Singh, Mishra & Nahar, 2002). A major part of the energy consumed by irrigation is supplied by non-renewable fossil fuels (Mrini, Senhaji & Pimentel, 2001; Topak et al., 2005). Dependence on fossil fuels, which are widely used to generate energy, could pose a threat to global food production growth and stability (Deike, Pallutt & Christen, 2008; Pimentel et al., 2002). In the recent past, the water-energy nexus, particularly irrigation energy efficiency, has received due attention from researchers and policymakers (Barik et al., 2017; Badiani & Jessoe, 2019; Sarkar, 2020; Singh, Kasana & Bhardwaj, 2022). Since the introduction of diesel and electric engines in the mid-twentieth century, groundwater has been used extensively, resulting in its fast depletion (Scanlon et al., 2012). The widespread groundwater abstraction for irrigation in India is also stated to be coupled with access to subsidized or free electricity in the country (Rajan & Ghosh, 2019; Sarkar, 2020). Mushtaq et al. (2009) opined that implementing efficient water management practices will increase water productivity and reduce energy dependency.

Groundwater depletion in the north-western Indian states of Punjab and Haryana is primarily due to energy-intensive agricultural practices following the Green Revolution in the 1960s (Van Dijk et al., 2020). Haryana has a considerable area under wheat-rice crop rotation and produces considerably more wheat and rice than the national average in India (Directorate of Economics and Statistics (DES), 2016). Dependence on groundwater increased due to the limited availability of canal water to meet the irrigation water requirements of key cereal crops in state (Kaur, Sidhu & Vatta, 2010; Asoka et al., 2017). Consequently, share of groundwater in net irrigated area of Haryana has climbed from 22.7% in 1966–1967 to 63.0% in 2019–2020 (Department of Economic and Statistical Analysis, Government of Haryana, India, 2021), while net irrigated area in state has increased from 1.2 million ha in 1966–1967 to 3.26 million ha in 2019–2020 (Department of Economic and Statistical Analysis, Government of Haryana, India, 2020). The number of tube wells in Haryana also expanded from approximately 25,000 in 1966–1967 to about 790,000 in 2019–2020 (67% electric and 33% diesel sets), which are mainly owned privately by farmers and used to meet crop water needs.

The nation is already confronting issues in the interplay of water, energy, and food systems as a result of its fast-expanding population and rising energy consumption. Earlier research shows that groundwater use for irrigated agriculture has increased vastly over the past decades, and groundwater pumping consumed large amounts of electricity and diesel, significantly contributing to the country’s total carbon emissions (Mushtaq et al., 2009; Karimi et al., 2012). Investigating the performance of submersible pumps operating in different sites in Bangladesh, it was revealed that the efficiency of the new pumps in a field was reduced by 20–40% than lab test results (Haque et al., 2017). The main causes of lower efficiency were improper matching of pump standard conditions and operation/system requirements. Sharma & Gupta (2016) found during a field survey to study the performance of agricultural pumping systems, observed that 25% of pumps were found in very high, 45% in excessive, and 30% in an intermediate range of electricity consumption, whereas only 15% pumps were operating at greater than 60% efficiency. Analyzing the increase in groundwater and energy use for pumping aligns with India’s overall development strategy of achieving food security. India’s agriculture sector accounts for 17.5% of the country’s total electricity consumption (Central Electricity Authority of India (CEA), 2021). Total power consumption was recorded at 228,172 GWh in 2020, up from 4,470 GWh in 1971 to meet the country’s agriculture sector needs. There are around 21 million agricultural pumping sets connected to the electricity grid in India, which consume approximately 187 billion kilowatt-hours (kWh) of electricity annually. Additionally, there are nearly 0.25 to 0.5 million new pumping sets being installed each year. The overall efficiency of pump sets being utilized for irrigation purpose has been found to be dismally low (Manoharan et al., 2021). Existing inefficient pump sets have an average efficiency of 25–30%, but new star-rated energy-efficient pump sets (EEPS) have an average efficiency of 40–45% (International Energy Agency (IEA), 2009; Saini, 2013). Thus, there is a great scope of energy savings potential in the agriculture sector in India. Another pathway for energy saving in groundwater pumping could be reduced groundwater withdrawal and enhanced water application efficiency. However, it is important to note that electricity costs to farmers are not high enough in the region to incentivize the adoption of improved efficiency of irrigation. Moreover, there are no associated water costs/charges (for groundwater) that might incentivize water saving. In view of the above, it appears that under present circumstances, implementing mechanical measures (e.g., use of efficient pump sets, at least for the new installations) is more practicable than the management measures (e.g., improving water application efficiency). The current study was undertaken in a part of Haryana’s Sonipat district, where groundwater pumping supplies more than 80% of the net irrigated area. This study aims to quantify the current level of energy consumption, pumping efficiencies, and potential energy saving at the farmer field level.

Materials and Methods

Study area

Sonipat district lies between latitude of 28°48′15″N to 29°17′10″N and longitude of 76°28′40″E to 77°12′45″E (Fig. 1) covering an area of 2,260.53 km2. The Sonipat district is further divided into eight administrative blocks, viz. Sonipat, Rai, Murthal, Ganaur, Kharkhoda, Gohana, Mudlana, and Kathura. The present study was carried out in the Rai and Sonipat blocks of the Sonipat district. The area under the Rai and Sonipat blocks is 280.49 and 397.89 km2, respectively, having sandy loam to loamy sand soil. The total number of electric pumping sets in Rai and Sonipat blocks during the year 2018–2019 were 5,136 and 6,897 compared to 3,456 and 4,587 during 2011–2012, respectively, showing that the numbers of electric pumping sets in these blocks are continuously increasing.

Figure 1 Map of study area.

Sampling strategy

A total of 65 pumping sets (35 from Rai and 30 from Sonipat block) were selected randomly for the study. All were equipped with submersible pumps and were being operated by electrical power. The selected pumping sets operated at different water table depths using horsepower (HP) motors. The farmers’ irrigation practice was studied in detail at 10 of the selected 65 tube wells.

Energy assessment

The present study is based on data collected from farmers and actual parameters measured at the selected site during the period spanning from 2017 to 2020. All the information related to the pumping set, its owner and their landholding was recorded. For this, a questionnaire was prepared and used as a research tool for data collection per the study’s objectives.

In order to estimate the current level of energy consumption, pumping efficiencies, and potential saving of electricity consumption involved in groundwater pumping for irrigation, information/measurement on the following parameters as required: area of major crops under groundwater irrigation, actual and potential water application efficiency of the prevalent method of irrigation, net and gross irrigation requirement of the crops, actual and minimum expected efficiency of the pumping sets, groundwater depth and head losses in the pumping systems.

Efficiency of pumping sets

The overall efficiency of the tube well was computed as the ratio of estimated power consumption to meet water horsepower & the actual power consumption as under

(1) η=0.746×WHPActualmeasuredpowerconsumption(kWh)

where, η is the overall efficiency of the pumping set. The water horsepower (WHP) was computed based on the discharge of the tube well and the estimated total head.

All the selected tube wells were owned privately by the farmers, and no reliable record was available regarding the model/make of the pumping set and the design performance characteristics. The design/expected efficiency of the installed tube well, thus, was determined based on Bureau of India Standards (BIS) code IS 8034:2002. Knowing the number of stages for different pump sets (which was obtained from the farmers during the survey), the head per stage was determined by dividing the total head with the number of stages. Then, using the value of head per stage (m) and discharge (l s−1) for pump sets, minimum efficiency was determined from standard charts given in BIS code IS 8034:2002. The efficiency in charts represents efficiency for three or more stages. For single-stage and two-stage pumps, the efficiency was multiplied by a factor of 0.97 and 0.98, respectively, as per instructions contained in the code. The efficiency thus obtained was further multiplied by the motor efficiency factor given in the same code. As per the BIS chart, the overall minimum expected efficiency of the pump set was estimated by multiplying it by the corresponding factor based on the number of stages and the corresponding motor efficiency factor.

(2) ηmin.=ηchart×fn×fm

where, ηmin is minimum expected efficiency (%), ηchart is the expected efficiency from the chart based on head per stage and discharge, fn is the efficiency factor based on the number of stages, and fm is the motor efficiency factor. Values of both fn and fm were taken from BIS code IS 8034:2002. The minimum expected overall efficiency varied from 42.9% to 52.4% in the Sonipat block and 45.1% to 52.4% in the Rai block.

Irrigation water application efficiency

Measurements at 10 farmer’s fields (five in each block) having tube well irrigation were carried out for wheat crops during 2018–2019 to assess the actual water application efficiency in the study area. All the farmers were using the border method of surface irrigation, and the cropping pattern was predominantly Paddy-Wheat. Generally, wheat receives 4–5 post-sowing irrigation in the study area. Knowing the bulk density, field capacity, and actual soil moisture measured in the root zone before irrigation, the required irrigation depth was computed at each selected farmer’s field (Michael, 2008). Applied depth of irrigation was computed for the first and fourth irrigation event based on the area of the border strip (Ab, m2), duration of irrigation (Ti, min), and discharge (Q, LPS) of tube wells, as given below.

Applieddepthofirrigation(mm)=60QTiAb(3)

Thereafter, the application efficiency was calculated as the ratio of required and applied irrigation depth for respective irrigations.

Waterapplicationefficiency(%)=Requireddepthofirrigation(mm)Applieddepthofirrigation(mm)×100(4)

Data pertaining to irrigation practices at ten selected farmer fields are given in Table 1. Based on these observations, the actual water application efficiency of irrigation in the study area was found to be 62.6%.

Table 1 Details of irrigation events during first (Ist) and fourth (IVth) irrigation to wheat crop at the fields of 10 selected farmers in the study area.

S. No.	Discharge of tube
well (l s−1)	Soil
texture	Required depth of
irrigation (cm)	Applied depth of
irrigation (cm)	Application
efficiency (%)	
Ist	IVth	Ist	IVth	Ist	IVth	
1	18.60	SL	6.21	6.11	10.42	10.09	59.59	60.55	
2	19.32	LS	6.28	6.14	9.79	9.28	64.14	66.16	
3	19.26	LS	6.39	6.26	10.45	9.59	61.14	65.27	
4	15.53	SL	5.97	5.78	9.39	9.25	63.57	62.48	
5	11.45	SL	5.94	5.84	8.65	8.55	68.67	68.30	
6	18.03	SL	6.11	5.92	9.78	9.30	62.47	63.65	
7	16.47	SL	5.96	5.76	9.66	9.37	61.69	61.47	
8	17.29	SL	6.21	6.03	9.84	9.68	63.10	62.29	
9	15.07	SL	6.27	6.20	10.32	10.18	60.75	60.90	
10	23.54	SL	6.31	6.13	10.88	10.67	57.99	57.45	
Mean	17.46	–	6.17	6.02	9.92	9.60	62.31	62.85	
Note:

SL, Sandy loam; LS, Loamy sand.

Energy requirement

Knowing the water requirement of different crops and pumping efficiency (existing as well as the minimum expected), the energy requirement was estimated by using the following relationship:

(5) E=27.27×H×WaterpumpedPumpingEfficiency

in which E is the estimated electricity energy consumed (kWh); H is the total head on the pump (m), which is the sum of the depth of pumping water level below the ground surface (Fig. 2), the height of delivery pipe above the ground surface and total head loss due to friction. The water pumped for a particular crop depends on the gross depth of irrigation. Water pumped (ha-m) was estimated as under:

Figure 2 Water table depth at the famers tubewells in the Sonipat and Rai blocks.

Waterpumped=Netirrigationrequirement(mm)10×Waterapplicationefficiency(6)

Net irrigation requirements for wheat (325 mm) and paddy (975 mm) were adopted as per the recommendations of Dhindwal, Phogat & Hooda (2008). As indicated earlier, the water application efficiency was taken as 62.6% (Singh, 2021).

Results and discussion

Overall and minimum expected efficiency

Electrical energy requirements for groundwater pumping in major crops grown in the Sonipat & Rai blocks were computed at the existing level of overall efficiency and based on minimum expected efficiency. The overall efficiencies of selected tube wells, as computed (Eq. (1)) based on actual measured power consumption, varied from 10.1% to 56.6% in the Sonipat block and 15.3% to 52.8% in the Rai block (Fig. 3). The difference between overall efficiency based on actual power consumption and expected minimum efficiency for different tube wells is also shown in Fig. 4. It can be seen that the actual overall efficiency of only two of the selected tube wells (TW-17 & TW-48) was higher than the minimum expected overall efficiency in the study area. The actual overall efficiency of TW-17 was noted as 56.6% against the minimum expected overall efficiency of 49.3% (i.e., 7.4% higher than the minimum expected efficiency). In comparison, the actual overall efficiency of TW-48 was noted as 52.8% against the minimum expected overall efficiency of 50.2%. The possible reason for the good efficiency of these tube wells may be that both the tubewells were newly installed at the time of observation, i.e., TW-17 (8-month-old) and TW-48 (3-month-old). For all other selected tube wells in the study area, the actual overall efficiency was lower by 4.5% to 38.7% from the respective minimum expected efficiency. It may be stated that most pump sets working in the study area are inefficient due to improper selection and poor maintenance of the pumps. The tubewell pumping sets in the study area were found to be working at an overall average actual efficiency of 26.1% & 32.5% against the minimum expected average efficiency of 47.6% & 49.7% in the Sonipat & Rai block, respectively. Further, it was also observed that some of the tube wells (e.g., 17 & 48) were found to be working at reasonably good efficiency, indicating that it is possible to bridge the gap between actual and expected minimum efficiency. Therefore, there is a lot of scope for energy conservation in the study area. Declining groundwater depth in the Rai block has resulted in increased energy demand over the years, and there is a need to improve water application efficiency and groundwater status in the study area.

Figure 3 Efficiency based on actual power consumption and expected minimum efficiency for the selected tube well in the Sonipat and Rai blocks.

Figure 4 Difference between overall efficiency based on actual power consumption and expected minimum efficiency for the selected tube well in the Sonipat and Rai blocks.

Pump sets in the study area were found to be working at an overall average actual efficiency of 26.1% & 32.5% in the Sonipat & Rai blocks, respectively. These results are inline with previous studies conducted in the region. World bank in an energy audit of electrical pump sets in Haryana found that average overall efficiency of pump sets varied between 21–24% and only 2% of the pump sets had average overall efficiency above 40% (World Bank, 2001). National productivity council (NPC) reported average overall efficiency of pumps in Haryana to be between 25% to 35% (National Productivity Council (NPC), Government of India (NPC), 2009). Another study suggested that average efficiency of pump sets in Haryana is 34.7% (Sood, 2010). Pilot studies conducted under the Agriculture Demand Side Management (Ag-DSM) scheme suggested that efficiencies can be increased from the existing level of 35% to 50% by advocating replacement of existing pump sets to Bureau of Energy Efficiency (BEE) labelled pump sets (Tyagi & Joshi, 2019).

Energy use and savings

The energy requirement for groundwater pumping in wheat and paddy crop at different farmers’ fields according to their irrigated area (Fig. 5) was computed using Eq. (5) and is given in Tables 2 and 3. Irrigated area cropped by selected farmers varies between 0.81–7.69 ha and 1.21–4.86 ha in Sonipat and Rai blocks, respectively. Energy requirement at the current efficiency level was 1,162.8–12,976.1 kWh and 3,490.7–38,953.3 kWh for wheat and paddy crops, respectively, in tube wells from the Sonipat block, while it was 1,469.5–6,975.8 kWh and 4,411.5–20,940.7 kWh for wheat and paddy crop, respectively in Rai block. The average energy requirement at the current efficiency level for both wheat (4,354.0 kWh) and paddy crop (13,100.4 kWh) in the Sonipat block were recorded higher than the corresponding value for wheat (3,424.8 kWh), and paddy (10,280.9 kWh) crop in Rai block as the average overall efficiency of tube wells working in Sonipat block (26.1%) was lower than Rai block (32.5%), while the average irrigated area cropped in both the blocks was almost equal (i.e., 2.56 and 2.57 ha) in the Sonipat and Rai blocks, respectively. Further, it may be noted that the energy requirement for paddy crop is higher than that for wheat crop due to its high-water requirement. Similar observations were realized by Khan, Khan & Latif (2010).

Figure 5 Irrigated area for selected farmers in the Sonipat and Rai blocks.

Table 2 Energy consumption based on actual and minimum expected efficiency in the Sonipat block.

	Energy consumption based on actual efficiency (kWh)	Energy consumption based on minimum expected efficiency (kWh)	
	Wheat	Paddy	Wheat	Paddy	
TW-1	4,503.4	13,518.9	2,823.6	8,476.1	
TW-2	4,013.9	12,049.5	2,016.8	6,054.3	
TW-3	5,115.9	15,357.5	2,130.3	6,395.0	
TW-4	3,358.9	10,083.1	1,369.3	4,110.7	
TW-5	5,319.4	15,968.4	2,971.7	8,920.7	
TW-6	1,912.4	5,740.9	1,212.0	3,638.2	
TW-7	12,976.1	38,953.3	7,004.6	21,027.4	
TW-8	7,167.2	21,515.5	4,184.2	12,560.6	
TW-9	7,216.1	21,662.1	3,008.7	9,032.0	
TW-10	2,566.2	7,703.5	1,532.1	4,599.1	
TW-11	3,204.2	9,618.8	1,913.0	5,742.6	
TW-12	7,517.0	22,565.4	1,769.6	5,312.3	
TW-13	3,127.4	9,388.2	1,462.7	4,390.8	
TW-14	11,059.2	33,198.9	2,288.9	6,871.1	
TW-15	1,517.0	4,553.8	903.7	2,713.0	
TW-16	2,336.8	7,014.9	640.7	1,923.3	
TW-17	1,789.5	5,371.8	2,056.5	6,173.5	
TW-18	8,921.3	26,781.1	4,634.4	13,912.3	
TW-19	6,355.8	19,079.6	2,367.8	7,108.1	
TW-20	1,584.0	4,755.2	758.2	2,276.0	
TW-21	5,094.6	15,293.6	3,426.7	10,286.6	
TW-22	2,843.1	8,534.7	1,844.4	5,536.8	
TW-23	2,888.1	8,669.9	1,953.6	5,864.6	
TW-24	4,679.5	14,047.5	2,025.8	6,081.2	
TW-25	1,162.8	3,490.7	566.4	1,700.3	
TW-26	1,772.7	5,321.4	1,046.4	3,141.4	
TW-27	4,991.2	14,983.4	3,036.2	9,114.4	
TW-28	2,579.2	7,742.5	1,801.5	5,407.9	
TW-29	1,779.9	5,343.2	1,159.5	3,480.6	
TW-30	1,567.1	4,704.4	796.1	2,390.0	
Mean	4,364.0	13,100.4	2,156.8	6,474.7	

Table 3 Energy consumption based on actual and minimum expected efficiency in the Rai block.

	Energy consumption based on actual efficiency (kWh)	Energy consumption based on minimum expected efficiency (kWh)	
	Wheat	Paddy	Wheat	Paddy	
TW-31	3,640.6	10,928.8	2,138.6	6,419.8	
TW-32	3,523.0	10,575.8	1,321.6	3,967.3	
TW-33	5,210.9	15,642.9	1,522.7	4,571.0	
TW-34	3,154.0	9,468.1	2,439.7	7,323.8	
TW-35	2,777.6	8,338.2	1,522.5	4,570.6	
TW-36	3,453.2	10,366.3	2,353.2	7,064.2	
TW-37	3,436.0	10,314.7	2,339.4	7,022.6	
TW-38	3,821.9	11,473.0	3,482.1	10,453.0	
TW-39	2,209.6	6,633.0	1,666.0	5,001.1	
TW-40	2,386.8	7,164.9	1,419.1	4,260.0	
TW-41	2,542.1	7,631.2	1,742.4	5,230.6	
TW-42	3,737.2	11,218.8	3,274.3	9,829.2	
TW-43	1,469.5	4,411.5	1,085.5	3,258.5	
TW-44	4,537.1	13,620.2	3,841.6	11,532.3	
TW-45	4,180.2	12,548.7	2,795.0	8,390.5	
TW-46	6,975.8	20,940.7	3,339.8	10,025.7	
TW-47	6,738.5	20,228.5	4,577.4	13,741.1	
TW-48	3,392.0	10,182.5	3,570.5	10,718.4	
TW-49	1,945.5	5,840.2	969.7	2,910.8	
TW-50	1,828.4	5,488.6	909.2	2,729.4	
TW-51	1,917.9	5,757.3	1,380.9	4,145.5	
TW-52	3,708.8	11,133.4	2,782.9	8,354.1	
TW-53	1,658.0	4,977.2	1,502.6	4,510.6	
TW-54	4,848.3	14,554.2	3,361.0	10,089.6	
TW-55	3,130.9	9,398.6	2,186.4	6,563.3	
TW-56	2,945.5	8,842.2	1,462.2	4,389.4	
TW-57	5,466.0	16,408.4	2,482.4	7,452.1	
TW-58	2,111.1	6,337.4	1,205.1	3,617.6	
TW-59	4,811.5	14,443.7	2,230.3	6,695.2	
TW-60	2,519.9	7,564.6	1,549.0	4,649.9	
TW-61	3,399.4	10,204.9	2,825.5	8,482.0	
TW-62	2,195.4	6,590.4	1,340.5	4,024.0	
TW-63	2,161.7	6,489.3	1,478.2	4,437.4	
TW-64	3,620.1	10,867.3	2,197.3	6,596.2	
TW-65	4,412.6	13,246.2	2,385.2	7,160.1	
Mean	3,424.8	10,280.9	2,190.8	6,576.8	

One of the possible interventions to reduce the pumping energy requirement for the existing cropping pattern and irrigation practice is to improve the efficiency of the pumping sets. As can be noted from Tables 2 and 3, the average energy requirement can be reduced by improving overall efficiency to the minimum expected efficiency level resulting in 566.4–7,004.6 kWh and 1,700.3–21,027.4 kWh energy use for wheat and paddy crops, respectively, in Sonipat block (Table 2) and 909.2–4,577.4 kWh and 2,729.4–13,741.1 kWh for wheat and paddy crop, respectively in the Rai block (Table 3). Energy saving potential for different tubewells in Sonipat and Rai blocks through improved efficiency of pump sets for wheat and paddy crops is given in Fig. 6. Energy saving by improving overall efficiency from current level to minimum expected level varied between 30.2–79.3% and 8.9–70.8% in Sonipat and Rai block, respectively (Fig. 7).

Figure 6 Energy saving (kWh) in Paddy-Wheat cropping system for the selected tube well in the Sonipat and Rai blocks.

Figure 7 Energy saving (%) in Paddy-Wheat cropping system for the selected tube well in the Sonipat and Rai blocks.

Average energy use per hectare was 1,761.9 and 5,288.9 kWh/ha for wheat and paddy crops, respectively, in the Sonipat block. Similarly, this was 1,370.2 kWh/ha and 4,113.1 kWh/ha for wheat and paddy crops, respectively, in Rai block (Fig. 8). Energy savings of 913.3 and 2,741.5 kWh/ha can be realized in wheat and paddy crop, respectively by improving the overall efficiency of pumps from current level to the minimum expected level in Sonipat block. Similarly, for the Rai block, 518.0 kWh/ha and 1,555.1 kWh/ha energy savings can be realized in wheat and paddy crop, respectively.

Figure 8 Energy use (kWh/ha) in Paddy-Wheat cropping system for the selected tube well in the Sonipat and Rai blocks.

Pump sets in the study area were working at lower overall efficiency of 26.1% & 32.5% against the minimum expected average efficiency of 47.6% & 49.7% in the Sonipat & Rai block, respectively. This suggests that replacement of exiting pumps with more efficiency pumps can be a viable option in the study area, leading to average energy savings of 48.3% and 35.9% in the Sonipat and Rai blocks, respectively. Replacement of existing irrigation pump sets with BEE/BIS approved ones resulted in energy saving of 23–37% during a pilot scale study conducted in Karnataka (Ravindranath, 2016). Similarly, 25% energy saving was realised by replacing about 2,209 inefficient pump sets from Mangalwedha subdivision of Solapur circle in Maharashtra (Nargundkar, 2017).

Efficiency can further be increased by proper design and installation of pump set according to pump characteristics curves, reducing other losses such as hydraulic losses, leakage losses, mechanical losses and disc friction losses in submersible pump set. Along with that other measures like improving cropping pattern by reallocation of crops and introduction of new less water intensive high yielding crops, accompanied by introducing water and energy saving technologies (like zero tillage, laser land levelling, and micro-irrigation, etc.) and ground water recharge can lead to substantial water and energy saving in the region.

Impact of groundwater level on energy use

It is evident from these results that there is great potential for energy saving in the study area by replacing the existing pump sets with suitable, efficient pump sets. However, the above intervention will be difficult to implement in the Rai block, where the water table is declining at a relatively rapid rate as compared to that in the Sonipat block (Fig. 9). Under declining groundwater conditions, properly selected pumps according to the groundwater depth at the time of installation may soon become less efficient due to the more rapidly changing groundwater depth with time. Therefore, in Rai block, until and unless suitable measures to control the decline in groundwater depth are undertaken (such as groundwater recharge and crop diversification towards less water-demanding crops), it may be very expensive to frequently replace the pumping sets to suit the resulting head on the pump due to declining groundwater depth. However, in the Sonipat block, where groundwater is not declining very rapidly, immediate replacement of inefficient pump sets with properly selected and efficient pump sets may be attempted to achieve the potential energy saving in groundwater pumping.

Figure 9 Ground water level during the year 1990–2020.

The estimated energy requirement in the Rai block for wheat and paddy crops considering the water table depth in 1990 and 2020, while other aspects kept the same, is shown in Fig. 10. The decline in groundwater depth has led to a considerable increase in energy demand in the Rai block. It can be clearly seen that if suitable groundwater recharge interventions can restore the groundwater level, the energy demand for different crops can be reduced considerably in the Rai block. For instance, the current energy demand for wheat crops in the Rai block based on a groundwater depth of 15.97 m is 11,059.1 MWh (Megawatt hour) and this would reduce to 6,696.6 MWh if groundwater depth can be restored to 1,990 level, i.e., 6.49 m depth. Similarly, the energy requirement for paddy crops can be reduced from 35,220.4 to 21,327.0 MWh by improving groundwater depth.

Figure 10 Energy use (kWh/ha) in wheat and paddy crops for the Rai block considering groundwater depth of year 1990 and 2020.

The groundwater table is an important factor affecting irrigation energy consumption. Most irrigated regions in the world are facing the problem of a continuous decline in groundwater, leading to an increase in irrigation energy consumption. The decline in groundwater level has increased the energy required for pumping the groundwater in many parts of India (Shah, 2009; Dhillon et al., 2018). Zhao et al. (2020) also attributed the decline of the groundwater table to an increase in energy consumption for pumping groundwater. It was also found that the decline of the groundwater table not only offset the effect of energy and water-saving measures but also consumed an additional 98.8 billion kWh of electric energy in the Hebei Province of China.

Most of the tube wells in the study area are already using low-friction HDPE pipes (High-Density Poly-Ethylene). However, in some of the tube wells, it was observed that head loss due to friction was quite high due to the use of a relatively small diameter column pipe. Therefore, in such tube wells, replacement of existing column pipe with large diameter column pipe may help to reduce the energy consumption. Likewise, the height of the delivery pipe may also be reduced to save energy consumption to a certain extent in the study area. Consumption of excessively higher energy by most of the pump sets compared to expected energy consumptions based on installed pump HP in the study area indicates that poor selection/maintenance of pump sets contributes a lot towards wasteful use of energy for the purpose of groundwater pumping. Non-availability of any authorized service centre for submersible motor and pumping sets was another major reason behind the lower efficiency of the tube wells. Farmers generally get repair work from local mechanics like rewinding & bush maintenance. The local mechanic has many limitations, like lack of proper training, machines, and tools. Also, they do not use authorized parts from reputed manufacturers and good quality materials for repairing these motors and pumps, which adversely affects the performance and efficiency of pumps.

This study has clearly indicated the need and scope of enhanced pumping efficiency of the existing tube wells in the study area. However, due to limited observation/information available under field conditions, it was not possible to identify the most effective strategy to be implemented to realise enhanced pumping efficiency of the tub wells. For instance, it was not possible to point out, on case to case basis, the actual rectification strategy to be adopted to achieve the desired level of pumping efficiency. For instance, in some of the cases it might be possible to achieved improved pumping efficiency through improved quality of pump repair and maintenance. On the other hand, in some of the cases, it might require the replacement of poorly selected pump set with a properly selected new pump set. Further there is need to implement suitable rectification measures on a pilot scale to demonstrate the actual energy saving potential for groundwater pumping.

Conclusions

The low overall efficiency of the existing tube wells (10–56.6%) indicated a wasteful use of energy for groundwater pumping for irrigation in the study area. Considering the scarcity of energy as well as recent concerns of energy generation related carbon emissions, it is imperative to take suitable measures for improving the efficiency of the pump sets.

Improving overall efficiency from the current level to the minimum expected level can lead to energy savings of 48.3% and 35.9% for tube wells in the Sonipat and Rai block, respectively. This suggests that adopting suitable corrective measures (e.g., proper selection and maintenance of pump sets) can lead to significant energy conservation.

In areas where groundwater is not declining rapidly (e.g., Sonipat block), replacement of existing inefficient pump sets with properly selected, and efficient pump sets may be attempted to achieve the potential energy saving in groundwater pumping. However, improving the efficiency of tube wells by replacing the existing inefficient pump sets alone may not be a sustainable solution in areas where groundwater is declining continuously (e.g., Rai block) and the same should be done in tandem with crop diversification, improving water application efficiency and groundwater status by means of improved irrigation management practices and adopting groundwater recharge techniques.

Supplemental Information

Supplemental Information 1 Raw data.

Click here for additional data file.

This research was part of a Ph.D. dissertation submitted by the first author to Chaudhary Charan Singh Haryana Agricultural University, Hisar, India.

Additional Information and Declarations

Competing Interests

Author Contributions

Data Availability

The authors declare that they have no competing interests.

Kuldeep Singh conceived and designed the experiments, performed the experiments, analyzed the data, prepared figures and/or tables, authored or reviewed drafts of the article, and approved the final draft.

Raj Kumar Jhorar conceived and designed the experiments, performed the experiments, analyzed the data, prepared figures and/or tables, authored or reviewed drafts of the article, and approved the final draft.

Manohar Sahai Sidhpuria conceived and designed the experiments, performed the experiments, authored or reviewed drafts of the article, and approved the final draft.

Mukesh Kumar conceived and designed the experiments, authored or reviewed drafts of the article, and approved the final draft.

Mukesh Kumar Mehla analyzed the data, prepared figures and/or tables, authored or reviewed drafts of the article, and approved the final draft.

The following information was supplied regarding data availability:

The raw measurements are available in the Supplemental File.

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
