# Peer review of "Energy conservation prospects in water intensive Paddy-Wheat cropping system for groundwater pumping in the semi-arid region of Haryana"

_PeerJ, doi:10.7717/peerj.14815_

## Round 0.1 · original submission · Major Revisions

Dear Authors,

The reviewers have now commented on your manuscript, as you shall see that they are suggesting a major revision. One of the main suggestions from their reviews is the issues with the basic reporting of results and rigorously strengthening discussion. Therefore it is requested to kindly go through the attached comments of the reviewers and revise your manuscript accordingly.

I shall be looking forward to receiving your manuscript in time.

Best regards
Gowhar Meraj

Reviewer 1 ·

Basic reporting

The manuscript has done an interesting job of investigating the contribution of the efficiency of the two regional pumps to irrigation energy efficiency, but I think the current state of the manuscript is still a big gap for publication. As there are too few results in this manuscript, which is the main body of the paper. The author only compares the energy consumption under the current efficiency of the pump and the assumed expected efficiency, which makes the content very thin. In the introduction section, the author introduces the situation of the two research areas in a large length, but does not review the current research progress and main conclusions on the relationship between water and energy in irrigation, and does not propose the innovation points and promotion value of this paper. In addition to improving the efficiency of the pump, it is suggested that the paper also compares and analyzes the energy saving prospect of reducing the amount of the irrigation water, enriching the content of the manuscript. Therefore, I recommend that the manuscript be major revised before acceptance.

Experimental design

Introduction
1 Line 50-51 “particularly irrigation energy efficiency, has received little attention” In fact, there is a lot of literatures on this, which is the current hot spot, should be cited
2 The introduction is not well written. It should supplement the current research progress and main conclusions of water-energy coupling in irrigation, and what are the innovation points of this study. It only introduces the situation of the research area, which is more like a report than an article that can be spread

Materials & Methods
3 L102-103, which data are obtained through questionnaires, how to avoid subjective errors in data
4 L117,How 0.746 is obtained, what is the reference, and why is this value used to represent pumping for the entire region
5 L134 How the values of fn and fn are obtained
6 L147 Why only use the first and fourth irrigation event
7 L162 What does 27.27 stand for?
8 L170 What does “10”mean in the formula

Validity of the findings

Results
9 L194 The drop of groundwater table and the efficiency of pump are two different factors. The influence of groundwater table variation on energy consumption is not analyzed in this manuscript. It is suggested to increase the study and analysis of groundwater table variation on energy consumption
10 The manuscript compared to the actual operation of the pump efficiency and the difference between the expected efficiency, as the prospect of the energy saving irrigation, in fact the “prospect” is very difficult to realize, on the one hand, pump actual operation condition is very complex, it is difficult to meet the efficiency curve of the factory. Even if the cost is not considered to replace the new pump is difficult to achieve the expected operation efficiency. On the other hand, compared with improving the efficiency of the pump, saving water and reducing the amount of irrigation water may be a more feasible way to achieve energy saving effect. The so-called “prospect” of the manuscript does not include these contents.

Discussion
11 L211-228 This part can be put into Result to analyze the efficiency improvement of the pump and the contribution to energy saving of different crops and different regions
12 L233 It is mentioned in the manuscript that as the groundwater table drops, the pump efficiency will also decrease. At what water table is the expected pump efficiency in this study, and is it consistent with the actual pump operating water table

Reviewer 2 ·

Basic reporting

The manuscript clearly described the current level of energy consumption, pumping efficiencies, and potential energy saving at the farmer field level. Great job!

Experimental design

No comment

Validity of the findings

No comment

Additional comments

No comment

Reviewer 3 ·

Basic reporting

Dear Editor,
I have gone through the assigned manuscript "Energy conservation prospects in water intensive Paddy-Wheat cropping system for groundwater pumping in the semi-arid region of India" in which authors have produced many positive and welcome outcomes. This is an interesting case study which is highly significant for farming community, researchers and policy makers. Although this paper would benefit from some closer proofreading. It includes many linguistic problems that, at times make it difficult to follow. Some Specific comments are mentioned below, which can improve the quality standard of the MS. This manuscript can be considered for publication in peer J after major revisions.

Experimental design

In material and method section, proper experimental design is missing; sampling strategy for site selection must be defined.

Validity of the findings

specific comments
1. In abstract, methodology is not clear, most of the proportion is reflection result section it should reflect significant findings justifying your research hypothesis. State clearly the principal conclusion.

2. In each section (for example, Abstract, discussion and conclusion), abbreviations and acronyms must be explained and placed in parentheses only the first time they are used (ex.GI, PVC, RPVC, HP, HDPE etc).
3. Overall the discussion section can be improved significantly. A more cirtical discussion is warranted involving what should be the implications for improving irrigation efficiency for rice wheat cropping systems. Expand this section and support discussion with relevant citation of current literature.
4. The conclusions section should not be a summary of your study. The conclusions section should illustrate the mechanistic links of findings obtained under the experiments and some concrete recommendations must be emerged as ecosystem specific not niche specific.
5. In conclusion section, information on selection of pump set, type of foot valve, type of irrigation pipes, column pipe diameter, height of the delivery pipe, motor or engine capacity, etc. must be compiled in the tabular form according to the groundwater depth. Table should be applied in nature and useful for the farming community.

General comments
1. Lines 46-48: Must be rephrased meaningfully.
2. Lines 54-56: I cannot understand what the authors are trying to communicate here, please rephrase it meaningfully.
3. Lines 59-62: Must be rephrased meaningfully.
4. Lines 207 -224 must be the part of result section.

---

## Round 0.2 · Major Revisions

Dear Authors,

The manuscript still lacks a solid discussion. A solid discussion refers to a discussion section organized as main findings, comparison with previous studies, key strengths, key limitations, and conclusions. The number of references is too small.

Therefore it is strongly suggested to improve the discussion section through cross-referencing with other published studies as well as prominently highlight the limitations of this work and the future scope.

I am looking forward to receiving your manuscript soon.

Best regards
Gowhar Meraj

Reviewer 3 ·

Basic reporting

Lines 53-54: font size typo error must be corrected before the final submission
Line 63-64: must be rephrased meaningfully
Line 90: “Nearly 2.5 and 5 lakh new pumping sets are built each year” rephrased meaning fully

Experimental design

Line 112: The term “Electric” must be as “electric”
Line 191: the unit Water pumped (ha-m/ha) may be rechecked
Lines 185-186: font size typo error in the formula may be corrected

Validity of the findings

no comment

Additional comments

The authors have addressed my and other reviewers' concerns properly and the quality of the manuscript has improved significantly. Although some minor comments need to be addressed further before the final submission.


As the research study is niche specific therefore title of the article can be modulated as “Energy conservation prospects in water intensive Paddy-Wheat cropping system for groundwater pumping in the semi-arid of Haryana”

---

## Round 0.3 · accepted · Accept

Dear Authors, thanks for incorporating all the suggestions made by me and the reviewers. I congratulate you on this very good work. Keep up the good work and serve humanity.